# Fast Yet Effective Graph Unlearning through Influence Analysis

## ABSTRACT

Recent evolving data privacy policies and regulations have led to increasing interest in the problem of removing information from a machine learning model. In this paper, we consider Graph Neural Networks (GNNs) as the target model, and study the problem of *edge unlearning in GNNs*, i.e., learning a new GNN model as if a specified set of edges never existed in the training graph. Despite its practical importance, the problem remains elusive due to the non-convexity nature of GNNs and the large scale of the input graph. Our main technical contribution is three-fold: 1) we cast the problem of fast edge unlearning as estimating the influence of the edges to be removed and eliminating the estimated influence from the original model in one-shot; 2) we design a computationally and memory efficient algorithm named EraEdge for edge influence estimation and unlearning; 3) under standard regularity conditions, we prove that EraEdge converges to the desired model. A comprehensive set of experiments on four prominent GNN models and three benchmark graph datasets demonstrate that EraEdge achieves significant speedup gains over retraining from scratch without sacrificing the model accuracy too much. The speedup is even more outstanding on large graphs. Furthermore, EraEdge witnesses significantly higher model accuracy than the existing GNN unlearning approaches.

## 1 INTRODUCTION

Recent legislation such as the General Data Protection Regulation (GDPR) (Regulation, 2018), the California Consumer Privacy Act (CCPA) (Pardau, 2018), and the Personal Information Protection and Electronic Documents Act (PIPEDA) (Parliament, 2000) requires companies to remove private user data upon request. This has prompted the discussion of "right to be forgotten" (Kwak et al., 2017), which entitles users to get more control over their data by deleting it from learned models. In case a company has already used the data collected from users to train their machine learning (ML) models, these models need to be manipulated accordingly to reflect data deletion requests.

In this paper, we consider Graph Neural Networks (GNNs) that receive frequent edge removal requests as our target ML model. For example, consider a social network graph collected from an online social network platform that witnesses frequent insertion/deletion of users (nodes) and/or change of social relations between users (edges). Some of these structural changes can be accompanied with users' withdrawal requests of their data. In this paper, we only consider the requests of removing social relations (edges). Then the owner of the platform is obligated by the laws to remove the effect of the requested edges, so that the GNN models trained on the graph do not "remember" their corresponding social interactions.

In general, a naive solution to deleting user data from a trained ML model is to retrain the model on the training data which excludes the samples to be removed. However, retraining a model from scratch can be prohibitively expensive, especially for complex ML models and large training data. To address this issue, numerous efforts (Mahadevan & Mathioudakis, 2021; Brophy & Lowd, 2021; Cauwenberghs & Poggio, 2000; Cao & Yang, 2015) have been spent on designing efficient unlearning methods that can remove the effect of some particular data samples without model retraining. One of the main challenges is how to estimate the effects of a given training sample on model parameters (Golatkar et al., 2021), which has led to research focusing on simpler convex learning problem such a linear/logistic regression (Mahadevan & Mathioudakis, 2021), random forests (Brophy & Lowd, 2021), support vector machines (Cauwenberghs & Poggio, 2000) and k-means clustering (Ginart et al., 2019), for which a theoretical analysis was established. Although there have

been some works on unlearning in deep neural networks (Golatkar et al., 2020a;b; 2021; Guo et al., 2020), very few works (Chen et al., 2022; Chien et al., 2022) have investigated efficient unlearning in GNNs. These works can be distinguished into two categories: exact and approximate GNN unlearning. GraphEraser (Chen et al., 2022) is an exact unlearning method that retrains the GNN model on the graph that excludes the to-be-removed edges in an efficient way. It follows the basic idea of Sharded, Isolated, Sliced, and Aggregated (SISA) method (Bourtoule et al., 2021) and splits the training graph into several disjoint shards and train each shard model separately. Upon receiving an unlearning request, the model provider retrains only the affected shard model. Despite its efficiency, partitioning training data into disjoint shards severely damages the graph structure and thus incurs significant loss of target model accuracy (will be shown in our empirical evaluation). On the other hand, *approximate GNN unlearning* returns a sanitized GNN model which is statistically indistinguishable from the retrained model. Certified graph unlearning (Chien et al., 2022) can provide a theoretical privacy guarantee of the approximate GNN unlearning. However, it only considers some simplified GNN architectures such as simple graph convolutions (SGC) and their generalized PageRank (GPR) extensions. We aim to design the efficient approximate unlearning solutions that are model-agnostic, i.e., without making any assumption of the nature and complexity of the model.

In this paper, we design an efficient edge unlearning algorithm named EraEdge which directly modifies the parameters of the pre-trained model in one shot to remove the *influence* of the requested edges from the model. By adapting the idea of treating removal of data points as upweighting these data points (Koh & Liang, 2017), we compute the influence of the requested edges on the model as the change in model parameters due to upweighting these edges. However, due to the aggregation function of GNN models, it is non-trivial to estimate the change on GNN parameters as removing an edge $e(v_i, v_j)$ could affect not only the neighbors of $v_i$ and $v_j$ but also on multi-hops. Thus we design a new influence derivation method that takes the aggregation effect of GNN models into consideration when estimating the change in parameters. We address several theoretical and practical challenges of influence derivation due to the non-convexity nature of GNNs.

To demonstrate the efficiency and effectiveness of EraEdge, we systematically represent the empirical trade-off space between *unlearning efficiency* (i.e., the time performance of unlearning, *model accuracy* (i.e., the quality of the unlearned model), and *unlearning efficacy* (i.e., the extent to which the unlearned model has forgotten the removed edges). Our results show that, first, while achieving similar model accuracy and unlearning efficacy as the retrained model, EraEdge is significantly faster than retraining. For example, it speeds up the training time by $5.03\times$ for GCN model on Cora dataset. The speedup is even more outstanding on larger graphs; it can be two orders of magnitude on CS graph which contains around 160K edges. Second, EraEdge outperforms GraphEraser (Chen et al., 2022) considerably in model accuracy. For example, EraEdge witnesses an increase of $50\%$ in model accuracy on Cora dataset compared to GraphEraser. Furthermore, EraEdge is much faster than GraphEraser especially on large graphs. For instance, EraEdge is $5.8\times$ faster than GraphEraser on CS dataset. Additionally, EraEdge outperforms certified graph unlearning (CGU) (Chien et al., 2022) significantly in terms of target model accuracy and unlearning efficacy, while it demonstrates comparable edge forgetting ability as CGU.

In summary, we made the following four main contributions: 1) We cast the problem of edge unlearning as estimating the *influence* of a set of edges on GNNs while taking the aggregation effects of GNN models into consideration; 2) We design EraEdge, a computationally and memory efficient algorithm that applies a one-shot update to the original model by removing the estimated influence of the removed edges from the model; 3) We address several theoretical and practical challenges of deriving edge influence, and prove that EraEdge converges to the desired model under standard regularity conditions; 4) We perform an extensive set of experiments on four prominent GNN models and three benchmark graph datasets, and demonstrate the efficiency and effectiveness of EraEdge.

## 2 GRAPH NEURAL NETWORK

Given a graph $G(V, E)$ that consists of a set of nodes $V$ and their edges $E$, the goal of a Graph Neural Network (GNN) model is to learn a representation vector $\vec{h}$ (embedding) for each node $v$ in $G$ that can be used in downstream tasks (e.g., node classification, link prediction).

A GNN model updates the node embeddings through aggregating its neighbors' representations. The embedding corresponding to each node $v_i \in V$ at layer $l$ is updated according to $v_i$'s graph

| Symbol | Meaning |
|---|---|
| $G(V, E)$ | Original graph. |
| $E_{\mathrm{UL}}$ | The set of edges to be removed. |
| $E \backslash E_{\mathrm{UL}}$ | Edges remained after removal of $E_{\mathrm{UL}}$. |
| $\mathcal{A}_{\mathrm{L}}$ | A GNN learning algorithm. |
| $\mathcal{A}_{\mathrm{UL}}$ | An unlearning algorithm. |
| $\theta_{\mathrm{OR}}$ | Parameters of $\mathcal{A}_{\mathrm{L}}$ trained over $G(V, E)$. |
| $\theta_{\mathrm{RE}}$ | Parameters of $\mathcal{A}_{\mathrm{L}}$ (re)trained over $G(V, E \backslash E_{\mathrm{UL}})$. |
| $\theta_{\mathrm{UL}}$ | Parameters of $\mathcal{A}_{\mathrm{L}}$ obtained by $\mathcal{A}_{\mathrm{UL}}$. |

Table 1: Notations

neighborhood (typically 1-hop neighborhood). This update operation can be expressed as follows:

$$H^{(l+1)} = \sigma(\mathsf{AGGREGATE}(A, H^{(l)}, \theta^{(l)})), \tag{1}$$

where $\sigma$ is an activation function, $A$ is the ajacency matrix of the given graph $G$, and $\theta^{(l)}$ denotes the trainable parameters as layer $l$. The initial embeddings at $\ell = 0$ are set to the input features for all the nodes, i.e., $H^{(0)} = X$.

Different GNN models use different $\mathsf{AGGREGATE}$ functions. In this paper, we consider four representative GNN models, namely Graph Convolutional Networks (GCN) (Kipf & Welling, 2017), GraphSAGE (Hamilton et al., 2018), graph attention networks (GAT) (Veličković et al., 2018), and Graph Isomorphism Network (GIN) (Xu et al., 2019). These models differ on their $\mathsf{AGGREGATE}$ functions. We ignore the details of their $\mathsf{AGGREGATE}$ functions as our unlearning methods are model agnostic, and thus are independent from these functions.

After $K$ iterations of message passing, a $\mathsf{Readout}$ function pools the node embeddings at the last layer and produce the final prediction results. The $\mathsf{Readout}$ function varies by the learning tasks. In this paper, we consider node classification as the learning task and the $\mathsf{Readout}$ function is a softmax function.

$$\hat{\mathbf{Y}} = \mathrm{softmax}(H^{(K)}\theta^{(K)}). \tag{2}$$

The final output of the target model for node $v$ is a vector of probabilities, each corresponding to the predicted probability (or posterior) that $v$ is assigned to a class. We consider cross entropy loss (Cox, 1958) which is the de-facto choice for classification tasks. In the following sections, we use $\mathcal{L}(\theta; v, E)$ to denote the loss on node $v$ for simplicity because only edges are directly manipulated.

## 3 FORMULATION OF EDGE UNLEARNING PROBLEM

Despite that GNNs are widely applicable to many fields, there are very few studies (Chen et al., 2022; Chien et al., 2022) on graph unlearning so far. In this section, we will formulate the definition of the edge unlearning problem. Table 1 lists the notations we use in the paper. In this paper, we only consider edge unlearning. We will discuss how to extend edge unlearning to handle node unlearning in Section 7.

Let $\mathcal{G}$ be the set of all graphs. In this paper, we only consider undirected graphs. Let $\Theta$ be the parameter space of the GNN models. A learning algorithm $\mathcal{A}_{\mathrm{L}}$ is a function that maps an instance $G(V, E) \in \mathcal{G}$ to a parameter $\theta \in \Theta$. Let $\theta_{\mathrm{OR}}$ be the parameters of $\mathcal{A}_{\mathrm{L}}$ trained on $G$. Any user can submit an edge unlearning request to remove specific edges from $G$. In practice, unlearning requests are often submitted sequentially. For efficiency, we assume these requests are processed in a batch. As the response to these requests, $\mathcal{A}_{\mathrm{L}}$ has to erase the impacts of these edges and produce an unlearned model. A straightforward approach is to retrain the model on $G(V, E \backslash E_{\mathrm{UL}})$ from scratch and obtain the model parameters $\theta_{\mathrm{RE}}$. However, due to the high computational cost of retraining, an alternative solution is to apply a *unlearning* process $\mathcal{A}_{\mathrm{UL}}$ that takes $E_{\mathrm{UL}}$ and $\theta_{\mathrm{OR}}$ as the input, and outputs the *unlearned model*.

The retrained and unlearned models should be sufficiently close and ideally identical. There are two types of notations in the literature that quantify the closeness of the retrained and unlearning models: (1) *both retraining and unlearning models are indistinguishable in the parameter space*, i.e., distributions of model parameters of both retraining and unlearning models are sufficiently

close, where the distance in two distributions can be measured by $\ell_2$ distance (Wu et al., 2020) and KL divergence (Golatkar et al., 2020b); (2) *both models are indistinguishable in the output space*, i.e., distributions of the learning outputs by both models are sufficiently close, where the distance between two output distributions can be measured by either test accuracy (Thudi et al., 2021) or the privacy leakage of membership inference attack launched on model outputs (Graves et al., 2021; Baumhauer et al., 2020). We argue that indistinguishably of the parameter space is not suitable for GNNs, due to their non-convex loss functions (Tarun et al., 2021), as small changes of the training data can cause large changes in GNN parameters. Therefore, in this paper, *we consider the indistinguishability of the output space between retrained and unlearned models as our unlearning notion.* Formally, we define the edge unlearning problem as follows:

**Definition 1** (**Edge Unlearning Problem**). *Given a graph $G(V, E)$, a set of edges $E_{\mathrm{UL}} \subset E$ that are requested to be removed from $G$, a graph learning algorithm $\mathcal{A}_{\mathrm{L}}$ and its readout function $f$, then an edge unlearning algorithm $\mathcal{A}_{\mathrm{UL}}$ should satisfy the following:*

$$P(f(\theta_{\mathrm{RE}})|G_{\mathrm{UL}}) \approx P(f(\theta_{\mathrm{UL}})|G_{\mathrm{UL}}), \tag{3}$$

*where $G_{\mathrm{UL}} = G(V, E\backslash E_{\mathrm{UL}})$, and $P(f(\theta)|G)$ denotes the distribution of possible outputs of the model (with parameters $\theta$) on $G$.*

The readout function $f$ varies for different learning tasks. In this paper, we consider the softmax function (Eqn. (2)) as the readout function. There are various choices to measure the similarity between the output softmax vectors. We consider Jensen–Shannon divergence (JSD) in our experiments.

## 4 MAIN ALGORITHM: EFFICIENT EDGE UNLEARNING

Given a graph $G(V, E)$ as input, one often finds a proper model represented by $\theta$ that fits the data by minimizing an empirical loss. In this paper, we consider cross-entropy loss (Cox, 1958) for node classification as our loss function. The **or**iginal model $\theta_{\mathrm{OR}}$ is optimized by the following:

$$\theta_{\mathrm{OR}} = \arg \min_{\theta} \frac{1}{|V|} \sum_{v \in V} \mathcal{L}(\theta; v, E). \tag{4}$$

Assuming a set of edges $E_{\mathrm{UL}}$ is deleted from $G$ and the new graph after this deletion is represented by $G_{\mathrm{UL}} = G(V, E\backslash E_{\mathrm{UL}})$, **re**training the model will give us a new model parameter $\theta_{\mathrm{RE}}$ on $G_{\mathrm{UL}}$:

$$\theta_{\mathrm{RE}} = \arg \min_{\theta} \frac{1}{|V|} \sum_{v \in V} \mathcal{L}(\theta; v, E\backslash E_{\mathrm{UL}}). \tag{5}$$

Figure 1 gives an overview of our unlearning solution named EraEdge. A major difficulty, as expected, is that obtaining $\theta_{\mathrm{RE}}$ is prohibitively slow for complex networks and large datasets. To overcome this difficulty, the aim of EraEdge is to identify an update to $\theta_{\mathrm{OR}}$ through an analogous *one-shot* ***un**learning update*:

$$\theta_{\mathrm{UL}} = \theta_{\mathrm{OR}} - I_{E_{\mathrm{UL}}}, \tag{6}$$

where $I_{E_{\mathrm{UL}}}$ is the *influence* of $E_{\mathrm{UL}}$ on the target model, i.e., the change on the model parameters by $E_{\mathrm{UL}}$. In general, $I_{E_{\mathrm{UL}}}$ is a $K \times d$ matrix, where $K$ is the number of parameters in $\theta_{\mathrm{OR}}$ (and both $\theta_{\mathrm{RE}}$ and $\theta_{\mathrm{UL}}$), and $d$ is the dimension of each parameter (i.e. embedding). This update can be interpreted from

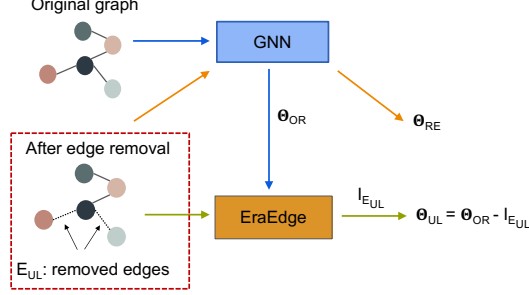

Figure 1: Framework of EraEdge. Orange lines indicate the process of retraining and green lines indicate unlearning.

the optimization perspective that the model forgets $E_{\mathrm{UL}}$ by "reversing" the influence of $E_{\mathrm{UL}}$ from the model. The challenge is how to quantify $I_{E_{\mathrm{UL}}}$ to achieve the unlearning objective (Eqn. (3)). Next, we discuss the details of how to compute $I_{E_{\mathrm{UL}}}$.

**Existing influence functions and their inapplicability.** Influence functions (Koh & Liang, 2017) enable efficient approximation of the effect of some particular training points on a model's prediction. The general idea of influence functions is the following: let $\theta$ and $\hat{\theta}$ be the model parameters before and after removing a data point $z$, the new parameters $\hat{\theta}_{\epsilon,z}$ after $z$ is removed can be computed as following:

$$\hat{\theta}_{\epsilon,z} = \arg\min_{\theta} \frac{1}{m} \sum_{z_i \neq z} \mathcal{L}(\theta; z_i) + \epsilon \mathcal{L}(\theta; z), \tag{7}$$

where $m$ is the number of data points in the original dataset, and $\epsilon$ is a small constant. Intuitively, the influence function computes the parameters after removal of $z$ by upweighting $z$ on the parameters with some small $\epsilon$.

It seemly sounds that the influence function (Eqn. (7)) can be applied to the edge unlearning setting directly by upweighting those nodes that are included in any edge in $E_{\mathrm{UL}}$. However, this is incorrect as removing one edge $e(v_i, v_j)$ from the graph can affect not only the prediction of $v_i$ and $v_j$ but also those of neighboring nodes of $v_i$ and $v_j$, due to the aggregation function of GNN models.

### 4.1 THEORETICAL CHARACTERIZATION OF EDGE INFLUENCE ON GNNS

In general, an $\ell$-layer GNN aggregates the information of the $\ell$-hop neighborhood of each node. Thus removing an edge $e(v_i, v_j)$ will affect not only $v_i$ and $v_j$ but also all nodes in the $\ell$-hop neighborhood of $v_i$ and $v_j$. To capture such aggregation effect in derivation of edge influence, first, we define the set of nodes (denoted as $V_e$) that will be affected by removing an edge $e(v_i, v_j)$ as:

$$V_e = \mathcal{N}(v_i) \cup \mathcal{N}(v_j) \cup \{v_i, v_j\},$$

where $\mathcal{N}(v)$ is the set of nodes connected to $v$ in $\ell$ hops. Then given a set of edges $E_{\mathrm{UL}} \subset E$ to be removed, the set of nodes $V_{E_{\mathrm{UL}}}$ that will be affected by removing $E_{\mathrm{UL}}$ is defined as follows:

$$V_{E_{\mathrm{UL}}} = \bigcup_{e \in E_{\mathrm{UL}}} V_e. \tag{8}$$

Next, we follow the data perturbation idea of influence functions (Koh & Liang, 2017), and compute the new parameters $\theta_{\epsilon, E_{\mathrm{UL}}}$ after the removal of $E_{\mathrm{UL}}$ as follows:

$$\theta_{\epsilon, V_{E_{\mathrm{UL}}}} = \arg\min_{\theta} \frac{1}{|V|} \sum_{v \in V} \mathcal{L}(\theta; v, E) + \epsilon \sum_{v \in V_{E_{\mathrm{UL}}}} \mathcal{L}(\theta; v, E \backslash E_{\mathrm{UL}}) - \epsilon \sum_{v \in V_{E_{\mathrm{UL}}}} \mathcal{L}(\theta; v, E). \tag{9}$$

Intuitively, Eqn. (9) approximates the effects that moving $\epsilon$ mass of perturbation on $V_{E_{\mathrm{UL}}}$ with $E \backslash E_{\mathrm{UL}}$ in place of $E$. Then we obtain the following theorem.

**Theorem 2.** *Given the parameters $\theta_{OR}$ obtained by $\mathcal{A}_{\mathrm{UL}}$ on a graph $G$, and the loss function $\mathcal{L}$, assume that $\mathcal{L}$ is twice-differentiable and convex in $\theta$, then the influence of a set of edges $E_{UL}$ is:*

$$I_{E_{\mathrm{UL}}} = -H_{\theta_{\mathrm{OR}}}^{-1} \left( \nabla_{\theta} \sum_{v \in V_{E_{\mathrm{UL}}}} \mathcal{L}(\theta_{\mathrm{OR}}; v, E \backslash E_{\mathrm{UL}}) - \nabla_{\theta} \sum_{v \in V_{E_{\mathrm{UL}}}} \mathcal{L}(\theta_{\mathrm{OR}}; v, E) \right) \tag{10}$$

*where $H_{OR} := \nabla^2 \frac{1}{|V|} \sum_{v \in V} L(\theta_{OR}, v, E)$ is the Hessian matrix of $L$ with respect to $\theta_{\mathrm{OR}}$.*

The proof of Theorem 2 can be found in Appendix A.1. According to Eqn (9), removing $E_{\mathrm{UL}}$ is equivalent to upweighting $\epsilon = \frac{1}{|V|}$ mass of perturbation. Therefore, $\theta_{\mathrm{UL}} = \theta_{\epsilon, V_{E_{\mathrm{UL}}}}$ when $\epsilon = \frac{1}{|V|}$. Finally, we have a linear approximation of $\theta_{\mathrm{UL}}$:

$$\theta_{\mathrm{UL}} \approx \theta_{\mathrm{OR}} + \frac{1}{|V|} I_{E_{\mathrm{UL}}}.$$

**Dealing with non-convexity of GNNs.** Theorem 2 assumes the loss function is convex. Given the non-convexity nature of GNN models, it is hard to reach the global minimum in practice. As a result, the Hessian matrix $H_{\theta_{\mathrm{OR}}}$ may have negative eigenvalues. To address this issue, we adapt the damping term based solution (Koh & Liang, 2017) to prevent $H_{\theta_{\mathrm{OR}}}$ from having negative eigenvalues by adding a damping term to the Hessian matrix, i.e., $(H_{\theta_{\mathrm{OR}}} + \lambda I)$.

## 4.2 Time and Memory Efficient Influence Estimator

Although by Theorem 2 estimating the edge influence amounts to solving a linear system, there are several practical and theoretical challenges. First, it can well be the case that the Hessian matrix $H_{\theta_{\mathrm{OR}}}$ is non-invertible. This is because our loss function is non-convex with respect to $\theta$. As a consequence, the linear system may even not have a solution. Second, even storing a Hessian matrix in memory (either CPU or GPU) is expensive: in our experiments, we will show that Hessian matrices are huge, e.g. the Hessian matrix on the Physics dataset has size around $10^6 \times 10^6$ which would cost 60 GB memory. Lastly, even under the promise that the linear system is feasible, computing the inverse of such a huge size matrix is prohibitive.

Our *second technical contribution* thus is an algorithm that resolves all the challenges mentioned above.

**Claim 3.** *There is a computationally and memory efficient algorithm to solve the linear system of $I_{E_{\mathrm{UL}}}$ in Theorem 2.*

The starting point of our algorithm is a novel perspective that solving the linear system (Eqn. (10)) can be thought of as finding a *stationary point* of the following quadratic function:

$$f(x) = \arg \min_x \frac{1}{2} x^T A x - b^T x, \tag{11}$$

with $A = H_{\theta_{\mathrm{OR}}}$ and $b = \nabla_\theta \sum_{v \in V_{E_{\mathrm{UL}}}} \mathcal{L}(\theta_{\mathrm{OR}}; v, E\backslash E_{\mathrm{UL}}) - \nabla_\theta \sum_{v \in V_{E_{\mathrm{UL}}}} \mathcal{L}(\theta_{\mathrm{OR}}; v, E)$. Note that even the function $f(x)$ is non-convex, there is rich literature establishing convergence guarantee to stationary points using gradient-descent-type algorithms; see e.g. (Bertsekas, 1999).

In this paper, we will employ the conjugate gradient (CG) method which exhibits promising computational efficiency for minimizing quadratic functions (Pytlak, 2008). In fact, it was well-known that as long as the step size satisfies the Wolfe conditions (Wolfe, 1969; 1971) and the objective function is Lipschitz and bounded from below, the sequence of iterates produced by CG asymptotically converges to a stationary point of $f(x)$, which corresponds to a solution $I_{E_{\mathrm{UL}}}$ that satisfies Eqn. (10). Note that these regularity conditions are satisfied as soon as the training data are bounded. Hence, we have the following convergence guarantee.

**Lemma 4** (Theorem 2.1 of (Pytlak, 2008)). *The CG method generates a sequence of iterates $\{x_t\}_{t \geq 1}$ such that $\lim_{t \to +\infty} f(x_t) = 0$. In addition, the per-iteration time complexity is $O(|x|)$ where $|x|$ denotes the dimension of $x$.*

We note, however, that an appealing feature of Eqn. (10) is that we do not have to find a solution with exact zero gradient. This enables us to terminate CG early by monitoring the magnitude of the gradients. Our empirical study also shows that CG can get good approximation in a small number of iterations.

In addition, we propose a memory-efficient implementation of CG, which significantly reduces the memory cost.

**Lemma 5.** *The CG method can be implemented using $O(|\theta|)$ memory.*

*Proof.* To see why the above lemma holds, recall that a key step of CG update is calculating the gradient of $f(x)$ as

$$\nabla f(x) = H_{\theta_{\mathrm{OR}}} x - \left( \nabla_\theta \sum_{v \in V_{E_{\mathrm{UL}}}} \mathcal{L}(\theta_{\mathrm{OR}}; v, E\backslash E_{\mathrm{UL}}) - \nabla_\theta \sum_{v \in V_{E_{\mathrm{UL}}}} \mathcal{L}(\theta_{\mathrm{OR}}; v, E) \right).$$

As $H_{\theta_{\mathrm{OR}}} \in \mathbb{R}^{|\theta| \times |\theta|}$, we can not explicitly compute $H_{\theta_{\mathrm{OR}}}$. Instead, we utilize Hessian-vector product (Pearlmutter, 1994) to approximately calculate $H_{\theta_{\mathrm{OR}}} x$ by

$$H_{\theta_{\mathrm{OR}}} x \approx \frac{g(\theta_{\mathrm{OR}} + rx) - g(\theta_{\mathrm{OR}})}{r}, \tag{12}$$

for some very small step size $r > 0$, where $g(\theta) := \nabla_\theta \sum_{v \in V_{E_{\mathrm{UL}}}} \mathcal{L}(\theta_{\mathrm{OR}}; v, E\backslash E_{\mathrm{UL}}) - \nabla_\theta \sum_{v \in V_{E_{\mathrm{UL}}}} \mathcal{L}(\theta_{\mathrm{OR}}; v, E)$. Note that the memory cost of evaluating the function value of $g(\cdot)$ is $O(|\theta|)$. Hence, Lemma 5 follows. $\square$

**Remark 6.** *Observe that a trivial implementation involves storing the Hessian matrix which consumes $O(|\theta|^2)$ memory. Returning to our previous example on the Physics dataset, a trivial implementation consumes 64 GB memory, while ours only needs 8 GB memory.*

*Proof of Claim 3.* Claim 3 follows from Lemma 4 and Lemma 5. □

## 5 EXPERIMENTS

In this section, we empirically verify the efficiency and effectiveness of our unlearning method.

### 5.1 EXPERIMENTAL SETUP

All the experiments are executed on a GPU server with NVIDIA A100 (40G). All the algorithms are implemented in Python with PyTorch. We set the damping term $\lambda = 0.01$ for all experiments. The link to the code and datasets will be available in the camera-ready version.

**Datasets.** We use three well-known datasets, namely **Cora** (Sen et al., 2008), **Citeseer** (Yang et al., 2016), and **CS** (Shchur et al., 2018), that are popularly used for performance evaluation of GNNs (Shchur et al., 2018; Zhang et al., 2019). The statistical information of these datasets can be found in Appendix A.2.

**GNN models.** We consider four representative GNN models, namely GCN (Kipf & Welling, 2017), GAT (Veličković et al., 2018), GraphSAGE (Hamilton et al., 2017), and GIN (Xu et al., 2019). We configure the GNNs with one hidden layer and a softmax output layer. All GNN models are trained for 1,000 epochs with an early-stopping condition when the validation loss is not decreasing for 20 epochs. We randomly split each graph into a training set (60%), a validation set (20%), and a test set (20%). As we mainly consider the impact of structure change on GNN models, we randomly initialize the values of node features such that they follow the Gaussian distribution to eliminate the possible dominant impact of node features on model performance. More details of the model setup can be found in Appendix A.2. We also measure the model performance with original node features. The results can be found in Appendix A.5.

**Picking edges for removal.** We randomly pick $k =\{100, 200, 400, 600, 800, 1,000\})$ edges from Cora and CiteSeer datasets, and $k=\{1,000, 2,000, 4,000, 6,000, 8,000, 10,000\})$ edges from CS dataset for removal. For each setting, we randomly sample ten batches of edges, with each batch containing $k$ edges. We report the average of model performance (model accuracy, unlearning efficacy, etc.) of the ten batches.

**Metrics.** We evaluate the performance of EraEdge in terms of *efficiency*, *efficacy*, and *model accuracy*: (1) **Unlearning efficiency**: we measure the running time of EraEdge and retraining time for a given set of edges; (2) **Target model accuracy**: we measure *accuracy* of node classification, i.e., the percentage of nodes that are correctly classified by the model, as the accuracy of the target model. Higher accuracy indicates better accuracy retained by the unlearned model; (3) **Unlearning efficacy**: we measure the distance between the output space of both retrained and unlearned models as the Jensen–Shannon divergence (JSD) between the posterior distributions output by these two models. Smaller JSD indicates a higher similarity between the two models in terms of their outputs.

**Baselines.** We consider both baselines of exact and approximate GNN unlearning for comparison with EraEdge. For exact GNN unlearning, we consider GraphEraser (Bourtoule et al., 2021) as the baseline. GraphEraser has two partitioning strategies denoted as *balanced LPA* (**BLPA**) and *balanced embedding k-means* (**BEKM**), We consider both BLPA and BEKM as the baseline methods. We use the same setting of number of shards as in (Chen et al., 2022) for both BLPA and BEKM. For approximate GNN unlearning, we consider (Chien et al., 2022) as the baseline.

### 5.2 PERFORMANCE OF EraEdge

We evaluate the performance of EraEdge on four representative GNN models and three graph datasets, and compare the performance of the unlearned model with both the retrained model and two baselines in terms of model accuracy, unlearning efficiency, and unlearning efficacy.

Table 2: **Performance of EraEdge, retrained models, and two baselines**. Different metrics are used to evaluate the performance, including the node classification accuracy of the retraining/unlearning model and running time. Each experiment is repeated 10 times. The results show that: (1) EraEdge is significantly faster than retraining while maintaining similar model accuracy; (2) EraEdge outperforms the baseline methods in model accuracy.

|  | # of Removed edges | Method | Accuracy (%) | Running time (s) |
|---|---|---|---|---|
| GCN + Cora | 200 | Retrain | $79.06 \pm 0.67$ | $3.83 \pm 0.44$ |
|  |  | BLPA | $48.06 \pm 2.14$ | $0.55 \pm 0.11$ |
|  |  | BEKM | $47.86 \pm 2.12$ | $0.56 \pm 0.11$ |
|  |  | EraEdge | $78.54 \pm 0.78$ | $0.90 \pm 0.18$ |
|  | 1,000 | Retrain | $75.24 \pm 1.22$ | $4.48 \pm 0.78$ |
|  |  | BLPA | $54.74 \pm 3.64$ | $0.81 \pm 0.15$ |
|  |  | BEKM | $54.80 \pm 3.60$ | $0.54 \pm 0.05$ |
|  |  | EraEdge | $74.76 \pm 1.73$ | $0.89 \pm 0.11$ |
| GraphSAGE + CS | 2,000 | Retrain | $87.19 \pm 0.22$ | $61.15 \pm 4.40$ |
|  |  | BLPA | $80.03 \pm 0.86$ | $5.38 \pm 1.30$ |
|  |  | BEKM | $80.80 \pm 4.90$ | $3.53 \pm 0.54$ |
|  |  | EraEdge | $87.14 \pm 0.21$ | $0.79 \pm 0.35$ |
|  | 10,000 | Retrain | $85.52 \pm 0.26$ | $57.13 \pm 3.51$ |
|  |  | BLPA | $80.25 \pm 0.76$ | $6.20 \pm 0.70$ |
|  |  | BEKM | $79.88 \pm 4.59$ | $4.01 \pm 0.36$ |
|  |  | EraEdge | $85.57 \pm 0.33$ | $0.91 \pm 0.43$ |

**Model accuracy.** We report the results of GNN model accuracy in Table 2 (*Accuracy* column) for GCN+Cora and GraphSAGE+CS settings. The results for other settings can be found in Appendix A.3. We have the following observations. First, the model accuracy obtained by EraEdge stays very close to that of the retrained model, regardless of the number of the removed edges. The difference in model accuracy between retrained and unlearned models remains negligible (in range of [0.48%, 0.52%] and [0.01%, 0.2%] for the two settings respectively). Second, EraEdge witnesses significantly higher model accuracy compared to the two baseline approaches, especially for the GCN+Cora setting. For example, both BEKM and BLPA only can deliver the model accuracy as around 48% when removing 200 edges under the GCN+Cora setting. This shows that unlearning through graph partitioning can bring significant loss of target model accuracy. Meanwhile EraEdge demonstrates that the model accuracy can be as high as ∼79% (65% improvement).

**Unlearning efficiency.** We report the time performance results of EraEdge and retraining in Table 2 (*Running time.* column) for GCN+Cora and GraphSAGE+CS settings. The results of other settings can be found in Appendix A.3. We measure the running time of the two baselines as the average training time per shard, as all shards are trained in parallel. The most important observation is that EraEdge is significantly faster than retraining. For example, it speeds up the training time by $5\times$ under GCN+Cora setting when removing 1,000 edges, and $77\times$ under GraphSAGE+CS setting when removing 2,000 edges. Furthermore, EraEdge is much faster than the two baselines especially when training large graphs. For example, EraEdge is $5.8\times$ faster than BLPA and $3.5\times$ faster than BEKM under the GraphSAGE+CS setting when 2,000 edges were removed.

**Unlearning efficacy.** Figure 2 plots the results of unlearning efficacy which is measured as the JSD between the posterior probability output by both retraining and unlearning models. We observe that JSD remains insignificant (at most 0.02) in all the settings. Furthermore, JSD stays relatively stable when the number of removed edges increase. This demonstrates the efficacy of EraEdge - it remains close to the retraining model even when a large number of edges is removed.

**Main takeaway.** While demonstrating similar accuracy as retraining, EraEdge is significantly faster than retraining, where the speedup gain becomes more outstanding when more edges are

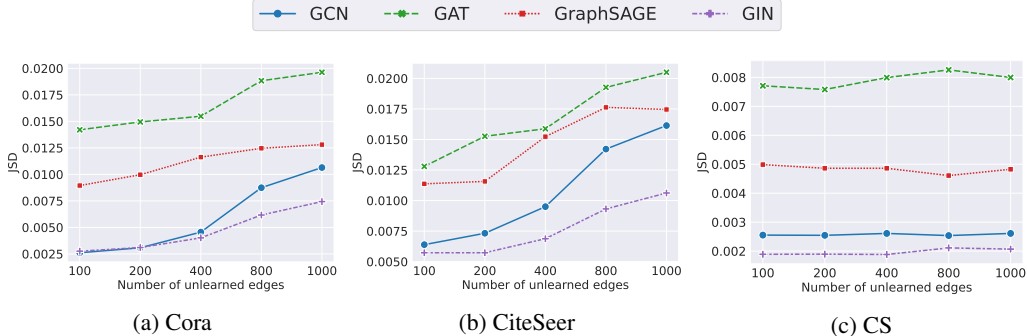

(a) Cora        (b) CiteSeer        (c) CS

Figure 2: **Unlearning efficacy**. We measure unlearning efficacy as the average JSD between the posterior distributions output by the retrained and unlearned models. Each experiment is repeated 10 times, where a different set of edges was sampled randomly for removal each time. The results show that the unlearned model is very close to the retrained model in the output space.

| $|E_{\mathrm{UL}}|$ | GCN | | | GAT | | |
|---|---|---|---|---|---|---|
| | Original | Retrain | EraEdge | Original | Retrain | EraEdge |
| 100 | 0.5913 | 0.5446 | 0.5297 | 0.6179 | 0.5615 | 0.5523 |
| 200 | 0.6014 | 0.5486 | 0.5471 | 0.5946 | 0.5659 | 0.5498 |
| 400 | 0.5978 | 0.5383 | 0.5378 | 0.5934 | 0.5400 | 0.5368 |
| 600 | 0.5993 | 0.5360 | 0.5383 | 0.6055 | 0.5471 | 0.5475 |
| 1000 | 0.5964 | 0.5399 | 0.5388 | 0.5983 | 0.5426 | 0.5441 |

Table 3: Edge forgetting ability of EraEdge measured as AUC of the membership inference attack that infers the existence of $E_{\mathrm{UL}}$ in training graph (Cora dataset)

removed. Furthermore, EraEdge outperforms the baseline approaches considerably in both model accuracy and time performance.

## 5.3 TESTING OF EDGE FORGETTING THROUGH MEMBERSHIP INFERENCE ATTACKS

To empirically evaluate the extent to which the unlearned model has forgotten the removed edges, we launch a black-box edge membership inference attack (MIA) (Wu et al., 2022)[1] that predicts whether particular edges exist in the training graph. We measure the attack performance as AUC of MIA. Intuitively, an AUC that is close to 50% indicates that MIA's belief of edge existence is close to random guess.

Table 3 reports the attack performance of MIA's inference of the removed edges $E_{\mathrm{UL}}$ against both the original model and the retrained/unlearned models on Cora dataset. First, MIA is effective to predict the existence of $E_{\mathrm{UL}}$ in the original graph, as the AUC of MIA against the original model is much higher than 0.5. Second, the ability of MIA inferring $E_{\mathrm{UL}}$ from either the retrained or the unlearned model degrades, as the AUC of MIA on both retrained and unlearned models is noticeably reduced. Indeed, the AUC of MIA for both retrained and unlearned models remain close to each other. This demonstrates that the extent to which EraEdge forgets $E_{\mathrm{UL}}$ is similar to that of the retrained model.

## 5.4 COMPARISON WITH CERTIFIED GRAPH UNLEARNING

In this part of the experiments, we compare the performance of EraEdge with certified graph unlearning (CGU) (Chien et al., 2022). The key idea of the certified unlearning method is to add noise drawn from the Gaussian distribution to the loss function. We use $\mu = 0$ and $\sigma = 1$ as the mean and standard deviation of the Gaussian distribution. We compare CGU and EraEdge in terms of: (1) target model accuracy, (2) unlearning efficacy (measured as the JSD between the probability output

---

[1]We use the implementation of LinkTeller available at: https://github.com/AI-secure/LinkTeller.

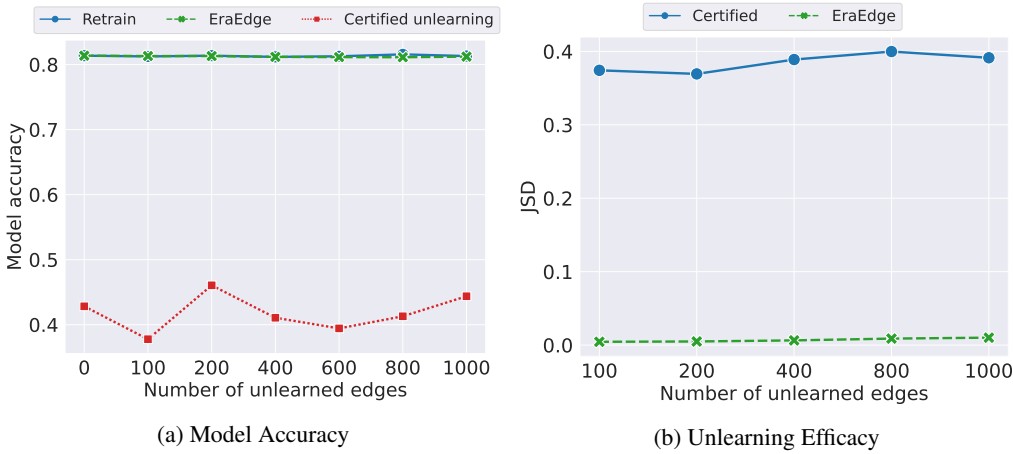

(a) Model Accuracy            (b) Unlearning Efficacy

Figure 3: Comparison between Certified Graph Unlearning (Chien et al., 2022) and Er-aEdge (GCN+Cora). We measure unlearning efficacy as the JSD between the distributions outputs by the retrained model and the unlearned model (either certified unlearning or EraEdge).

| $|E_{\mathrm{UL}}|$ | Original | Retrain | CGU | EraEdge |
|---|---|---|---|---|
| 100 | 0.5913 | 0.5446 | 0.5329 | 0.5297 |
| 200 | 0.6014 | 0.5486 | 0.5485 | 0.5471 |
| 400 | 0.5978 | 0.5383 | 0.5343 | 0.5378 |
| 600 | 0.5993 | 0.5360 | 0.5434 | 0.5383 |
| 1000 | 0.5964 | 0.5399 | 0.5379 | 0.5388 |

Table 4: Edge forgetting ability of both EraEdge and certified edge unlearning (CGU), where the ability is measured as AUC of the membership inference attack that infers existence of $E_{\mathrm{UL}}$ in training graph (GCN+Cora).

of retraining and unlearning models), and (3) privacy vulnerability of the removed edges against the membership inference attack.

Figure 3 (a) reports the target model accuracy by CGU and EraEdge. As we can see, while Er-aEdge enjoys similar target model accuracy as the retrained model, CGU suffers from significant loss of model accuracy due to added noise, where the model accuracy is 50% worse than that of both EraEdge and retraining.

Figure 3 (b) reports unlearning efficacy by CGU and EraEdge. The results demonstrate that the model output by CGU is much farther away from that of the retrained model than EraEdge. This is consistent with the low accuracy results in Figure 3 (b).

Table 4 shows the ability of forgetting the removed edges $E_{\mathrm{UL}}$ by both CGU and EraEdge, where the edge forgetting ability is measured as the accuracy (AUC) of the membership inference attack that predicts $E_{\mathrm{UL}}$ in the training graph. We use the same membership inference attack (Wu et al., 2022) as in Section 5.3. The reported results are calculated as the average AUC of ten MIA trials. We observe that CGU and EraEdge has comparable edge forgetting ability, where MIA performance against both models is close. This demonstrate empirically that EraEdge provides similar privacy risks as CGU. As it has been shown above that the target model accuracy by EraEdge out-performs that of CGU significantly, we believe that EraEdge better addresses the trade-off between unlearning efficacy, privacy, and model accuracy.

## 6 RELATED WORK

**Machine unlearning.** Machine unlearning aims to remove some specific information from a pre-trained ML model. Several attempts have been made to make unlearning more efficient than retrain-ing from scratch. An earlier study converts ML algorithms to statistical query (SQ) learning, so that

unlearning processes only need to retrain the summation of SQ learning (Cao & Yang, 2015). The concept of *SISA* (sharded, isolated, sliced, and aggregated) approach is proposed recently (Bourtoule et al., 2021) where a set of constituent models, trained on disjoint data *shards*, are aggregated to form an ensemble model. Given an unlearning request, only the affected constituent model is retrained. Alternative machine unlearning solutions directly modify the model's parameters to unlearn in a small number of updates (Guo et al., 2020; Neel et al., 2021; Sekhari et al., 2021). Recent studies have focused on various convex ML models including random forest (Brophy & Lowd, 2021; Schelter et al., 2021), k-means clustering (Ginart et al., 2019), and Bayesian inference models (Fu et al., 2021).

**Machine unlearning in deep neural networks.** Early work on deep machine unlearning focuses on removing the information from the network weights by imposing a condition of SGD based optimization during training (Golatkar et al., 2020a). The subsequent work (Golatkar et al., 2020b) estimates the network weights for the unlearned mode. However, all these methods suffer from high computational costs and constraints on the training process (Tarun et al., 2021). The *amnesiac unlearning* approach (Graves et al., 2021) focuses on Convolutional Neural Networks. It cancels parameter updates from only the batches containing the removed data. However, it assumes that the data to be removed is known before the training of the original model, which does not hold in our setting where edge removal requests are unknown and unpredictable. There also has been recent empirical and theoretical work in developing deep network unlearning in the application domain of computer vision (Du et al., 2019; Nguyen et al., 2020). *GraphEraser* (Chen et al., 2022) is one of the few works that consider unlearning in GNNs. It follows the SISA approach (Bourtoule et al., 2021) and splits graph into disjoint partitions (shards). Upon receiving an unlearning request, only the model on the affected shards is retrained. However, as splitting the training graph into disjoint partitions will damage the original graph structure, GraphEraser could downgrade the accuracy of the unlearned model significantly, especially when a large number of edges is to be removed. This has been demonstrated in our experiments.

**Certified machine unlearning.** *Certified removal* (Guo et al., 2020) defines approximate unlearning with a privacy guarantee (indistinguishability of unlearned models with retrained models), where indistinguishability is defined in a similar manner as differential privacy (Dwork et al., 2006). Certified removal can be realized by adding noise sampled from either Gaussian distribution or Laplace distribution on the weights (Golatkar et al., 2020a; Wu et al., 2020; Neel et al., 2021; Golatkar et al., 2021; Sekhari et al., 2021), or adding perturbation on the loss function (Guo et al., 2020). (Chien et al., 2022) provides the first certified GNN unlearning solution. It only considers simple graph convolutions (SGC) and their generalized PageRank (GPR) extensions. To achieve a theoretical guarantee for certified removal, it adds noise to the loss function. However, as shown in our empirical evaluation (Section 5), the certified unlearning leads to significant loss of target model due to the added noise.

**Explanations of deep ML models by influence functions.** One of the challenges of deep ML models is its non-transparency that hinders understanding of the prediction results. Recent works (Koh & Liang, 2017) adapt the concept of *influence function* - a classic technique from robust statistics — to formalize the impact of a training point on a prediction. Broadly speaking, the influence function attempts to estimate the change in the model's predictions if a particular training point is removed. Very recently, the concept of influence function has been extended to GNNs. For instance, influence functions are designed for GNNs to measure feature-label influence and label influence (Wang et al., 2019). Node-pair influence, i.e., the change in the prediction for node $u$ if the features of the other node $v$ are reweighted, is also studied (Wu et al., 2022). Unlike these works, we estimate the edge influence, i.e., the effect of removing particular edges on GNN models.

## 7 CONCLUSION

In this work, we study the problem of edge unlearning that aims to remove a set of target edges from GNNs. We design an approximate unlearning algorithm named EraEdge which enables fast yet effective edge unlearning in GNNs. An extensive set of experiments on four representative GNN models and three benchmark graph datasets demonstrates that EraEdge can achieve significant speedup gains over retraining without sacrificing the model accuracy too much.

There are several research directions for the future work. First, while EraEdge only considers edge unlearning, it can be easily extended to handle node unlearning, as removing a node $v$ from a graph

is equivalent to removing all the edges that connect with $v$ in the graph. We will investigate the feasibility and performance of node unlearning through EraEdge, and compare the performance with the existing node unlearning methods (Chien et al., 2022). Second, an important metric of unlearning performance is *unlearning capacity*, i.e., the maximum number of edges that can be deleted while still ensuring good model accuracy. We will investigate how EraEdge can be tuned to meet the capacity requirement. Third, we will extend the study to a relevant topic, *continual learning* (CL), which studies how to learn from an infinite stream of data, so that the acquired knowledge can be used for future learning (Chen & Liu, 2018). An interesting question is how to support both continual learning (Chen & Liu, 2018) and private unlearning (CLPU) (Liu et al., 2022), i.e., the model learns and remembers permanently the data samples at large, and forgets specific samples completely and privately. We will explore how to extend EraEdge to support CLPU.

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

| Dataset | #. Features | #. Nodes | #. Edges | #. Classes | Min Degree | Max Degree | Avg. Degree |
|---|---|---|---|---|---|---|---|
| **Cora** | 1,433 | 2,708 | 5,429 | 7 | 2 | 198 | 21.82 |
| **CiteSeer** | 3,703 | 3,327 | 4,552 | 6 | 2 | 126 | 13.83 |
| **CS** | 6,805 | 18,333 | 163,788 | 15 | 3 | 262 | 36.43 |

Table 5: Description of datasets

# A   APPENDIX

## A.1   PROOF OF THEOREM 2

*Proof.* For simplicity, we first define

$$R(\theta, V, E) = \sum_{v \in V} L(\theta, v, E). \tag{13}$$

Then, we formulate a GNN learning process as

$$\theta_{\mathrm{OR}} = \arg\min_{\theta} \frac{1}{|V|} R(\theta, V, E). \tag{14}$$

Since removing edges can be considered as perturbing the input, we introduce Eqn 9,

$$\theta_{\epsilon} = \arg\min_{\theta} \frac{1}{|V|} \sum_{v \in V} L(\theta; v, E) + \epsilon \sum_{v \in V_{E_{\mathrm{UL}}}} L(\theta; v, E \backslash E_{\mathrm{UL}}) - \epsilon \sum_{v \in V_{E_{\mathrm{UL}}}} L(\theta; v, E)$$

$$= \arg\min_{\theta} \frac{1}{|V|} R(\theta, V, E) + \epsilon R(\theta, V_{E_{\mathrm{UL}}}, E \backslash E_{\mathrm{UL}}) - \epsilon R(\theta, V_{E_{\mathrm{UL}}}, E). \tag{15}$$

We note a necessary condition is that the gradient of Eqn 15 at $\theta_{\epsilon}$ is zero. Then, we have

$$0 = \frac{1}{|V|} \nabla_{\theta} R(\theta_{\epsilon}, V, E) + \epsilon \nabla_{\theta} R(\theta_{\epsilon}, V_{E_{\mathrm{UL}}}, E \backslash E_{\mathrm{UL}}) - \epsilon \nabla_{\theta} R(\theta_{\epsilon}, V_{E_{\mathrm{UL}}}, E). \tag{16}$$

Next, we apply Taylor series at $\theta_{\mathrm{OR}}$ and we get

$$0 \approx \frac{1}{|V|} \nabla_{\theta} R(\theta_{\mathrm{OR}}, V, E) + \epsilon \nabla_{\theta} R(\theta_{\mathrm{OR}}, V_{E_{\mathrm{UL}}}, E \backslash E_{\mathrm{UL}}) - \epsilon \nabla_{\theta} R(\theta_{\mathrm{OR}}, V_{E_{\mathrm{UL}}}, E)$$

$$+ \left[ \frac{1}{|V|} \nabla_{\theta}^2 R(\theta_{\mathrm{OR}}, V, E) + \epsilon \nabla_{\theta}^2 R(\theta_{\mathrm{OR}}, V_{E_{\mathrm{UL}}}, E \backslash E_{\mathrm{UL}}) - \epsilon \nabla_{\theta}^2 R(\theta_{\mathrm{OR}}, V_{E_{\mathrm{UL}}}, E) \right] (\theta_{\epsilon} - \theta_{\mathrm{OR}}), \tag{17}$$

where we have dropped $o(\theta_{\mathrm{OR}} - \theta_{\epsilon})$ for approximation. Then Eqn (17) is a linear system of $E_{\mathrm{UL}}$, the influence of $E_{\mathrm{UL}}$. Since $\theta_{\mathrm{OR}}$ is the minimum of Eqn (14), we have $\frac{1}{|V|} \nabla R(\theta_{\mathrm{OR}}, V, E) = 0$. As $\epsilon$ is a small value, we drop the two $o(\epsilon)$ terms and have the following:

$$\frac{1}{|V|} \nabla_{\theta}^2 R(\theta_{\mathrm{OR}}, V, E)(\theta_{\epsilon} - \theta_{\mathrm{OR}}) + \epsilon \Big( \nabla_{\theta} R(\theta_{\mathrm{OR}}, V_{E_{\mathrm{UL}}}, E \backslash E_{\mathrm{UL}}) - \nabla_{\theta} R(\theta_{\mathrm{OR}}, V_{E_{\mathrm{UL}}}, E) \Big) \approx 0. \tag{18}$$

Suppose Eqn (14) is convex, then

$$\theta_{\epsilon} - \theta_{\mathrm{OR}} \approx -\frac{1}{|V|} \nabla_{\theta}^2 R(\theta_{\mathrm{OR}}, V, E)^{-1} \Big( \nabla_{\theta} R(\theta_{\mathrm{OR}}, V_{E_{\mathrm{UL}}}, E \backslash E_{\mathrm{UL}}) - \nabla_{\theta} R(\theta_{\mathrm{OR}}, V_{E_{\mathrm{UL}}}, E) \Big) \epsilon \tag{19}$$

Denote

$$I_{E_{\mathrm{UL}}} := \frac{d(\theta_{\epsilon} - \theta_{\mathrm{OR}})}{d\epsilon} \Big|_{\epsilon=0} = -H_{\theta_{\mathrm{OR}}}^{-1} \Big( \nabla_{\theta} R(\theta_{\mathrm{OR}}, V_{E_{\mathrm{UL}}}, E \backslash E_{\mathrm{UL}}) - \nabla_{\theta} R(\theta_{\mathrm{OR}}, V_{E_{\mathrm{UL}}}, E) \Big) \tag{20}$$

where $H_{OR} := \nabla^2 \frac{1}{|V|} \sum_{v \in V} L(\theta_{OR}, v, E)$. $\square$

| | Sequential unlearning | | | | Single-batch unlearning |
|---|---|---|---|---|---|
| | $B_1$ | $B_2$ | $B_3$ | $B_4$ | |
| Retrain | 0.7795 | 0.7758 | 0.7811 | 0.7797 | 0.7793 |
| EraEdge | 0.7791 | 0.7786 | 0.7790 | 0.7793 | 0.7792 |

Table 6: Target model accuracy under single-batch and sequential unlearning (GCN+Cora).

## A.2 ADDITIONAL DETAILS OF EXPERIMENTAL SETUP

**Description of datasets.** Table 5 summarizes the statistical information of the three graph datasets (Cora, Citeseer, and CS) we used in the experiments. Cora and Citeseer datasets are citation graphs, while CS dataset is a co-author graph.

**Additional details of model setup.** To ensure fair comparison between the retrained and unlearned models, we use the same model size (i.e., same number of layers and number of neurons) for both retraining and unlearned models. All GNN models are trained with a learning rate of 0.001. We train the models by 1,000 epochs, with the early-stopping condition as that the validation loss does not decrease for 20 epochs.

## A.3 ADDITIONAL PERFORMANCE RESULTS

**Model efficiency.** Figure 4 presents the model efficiency results on the three datasets. We observe that EraEdge is significantly faster than retraining. For example, EraEdge outperforms by $9.95\times$, $5.41\times$, $69.36\times$, and $3.12\times$ on CS dataset respectively over retraining (Figure 4 (c), (f), (i), and (l)).

**Model accuracy.** Figure 5 presents the results of model accuracy for all settings. First, the accuracy of the target model by EraEdge is very close to that by the retrained model. In particular, the average difference in model accuracy between retrained and unlearned models are in the range of [0.11%, 0.68%], [0.02%, 0.74%], [0.06%, 0.65%] and [0.07%, 1.00%] for GCN, GAT, GraphSAGE, and GIN on Cora, [0.01%, 0.71%], [0.05%, 0.44%], [0.05%, 0.65%], and [0.02%, 1.25%] on CiteSeer, and [0.02%, 0.22%], [0.01%, 0.20%, [0.05%, 0.23%], and [0.01%, 0.22%] on CS, respectively. Furthermore, the model accuracy of the unlearned model stays close to that of the retrained model, regardless of the number of removed edges. This demonstrates that EraEdge can handle the removal of a large number of edges.

## A.4 SEQUENTIAL UNLEARNING (NEW)

So far we only considered deleting of one batch of edges. In practice, there can be multiple batch deletion requests to forget the edges in a sequential fashion. Next, we focus on the scenario where multiple edge batches are removed sequentially. Specifically, we divide the to-be-removed $E_{\text{UL}}$ into $k > 1$ disjoint batches $\{B_i\}_{i=1}^k$, with each batch consisting of the same number of edges. For each batch $B_i$ ($1 \le i \le k-1$), we consider the target model obtained from retraining/unlearning of the previous batch $B_{i-1}$ as the original model $\theta_{\text{OR}}$, and update $\theta_{\text{OR}}$ by removing $B_i$ (either by retraining or unlearning). We evaluate the target model accuracy under sequential unlearning and compare it with that under one-batch unlearning.

We consider $k = 4$, and reports the target model accuracy for deleting $E_{\text{UL}}$ in one batch and deleting $E_{\text{UL}}$ in $k = 4$ batches in Table 6. We also report the target model accuracy of the retrained and unlearned models at each batch. We observe that, first, the accuracy of the unlearned model remains close to the retrained model at each batch during sequential removals. Second, the performance of the unlearned model after removing $k$ batches stays close to that of the model after single-batch unlearning. These results demonstrate that EraEdge can handle sequential deletion of multiple batches of edges.

## A.5 UNLEARNING WITH NODE FEATURES (NEW)

In Section 5 we mainly considered the node features that are randomly initialized to eliminate the possible dominant impact of node features on model performance. Next, we evaluate the perfor-

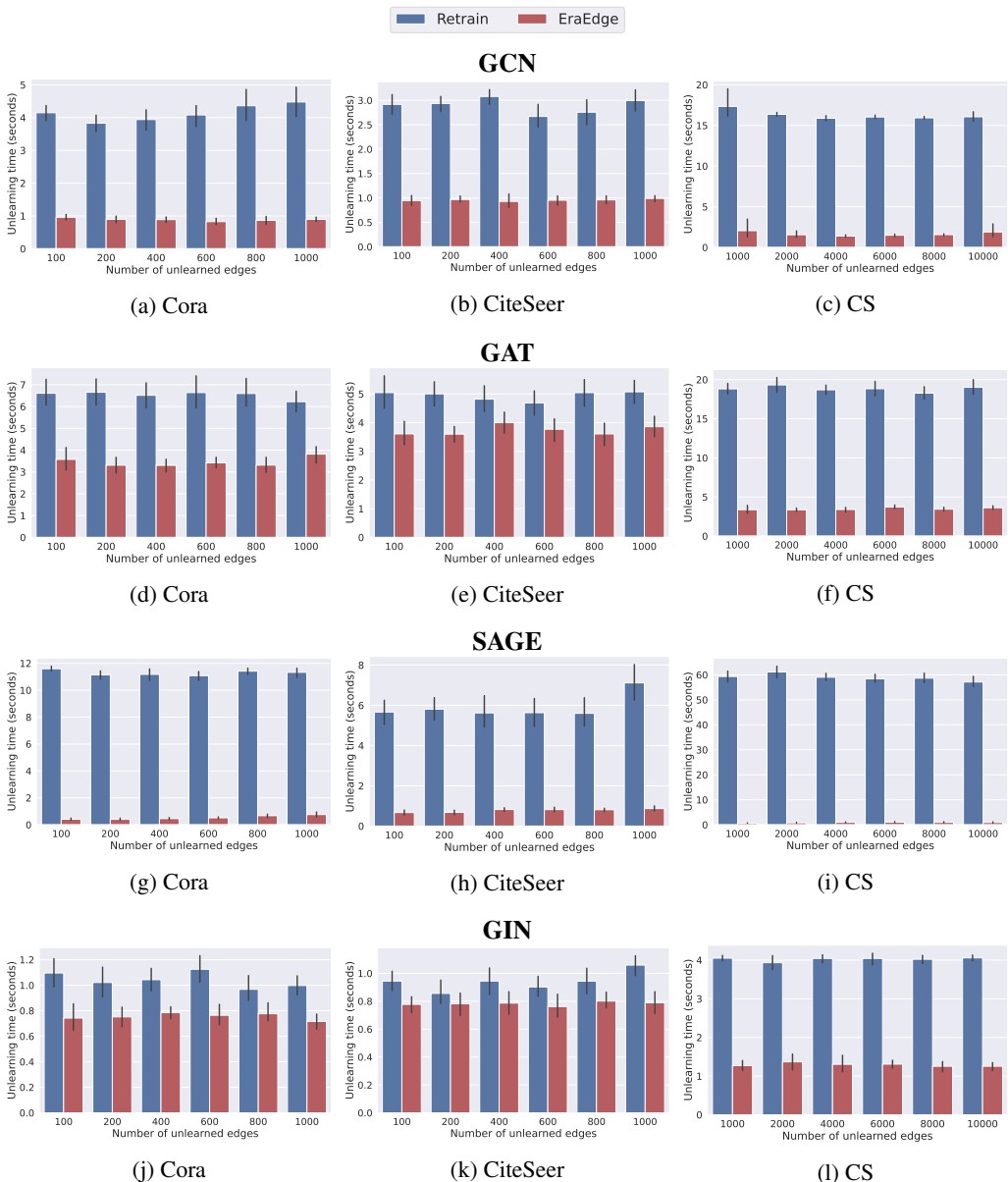

Figure 4: **Unlearning efficiency**. The results show that EraEdge speeds up the training time significantly compared with retraining.

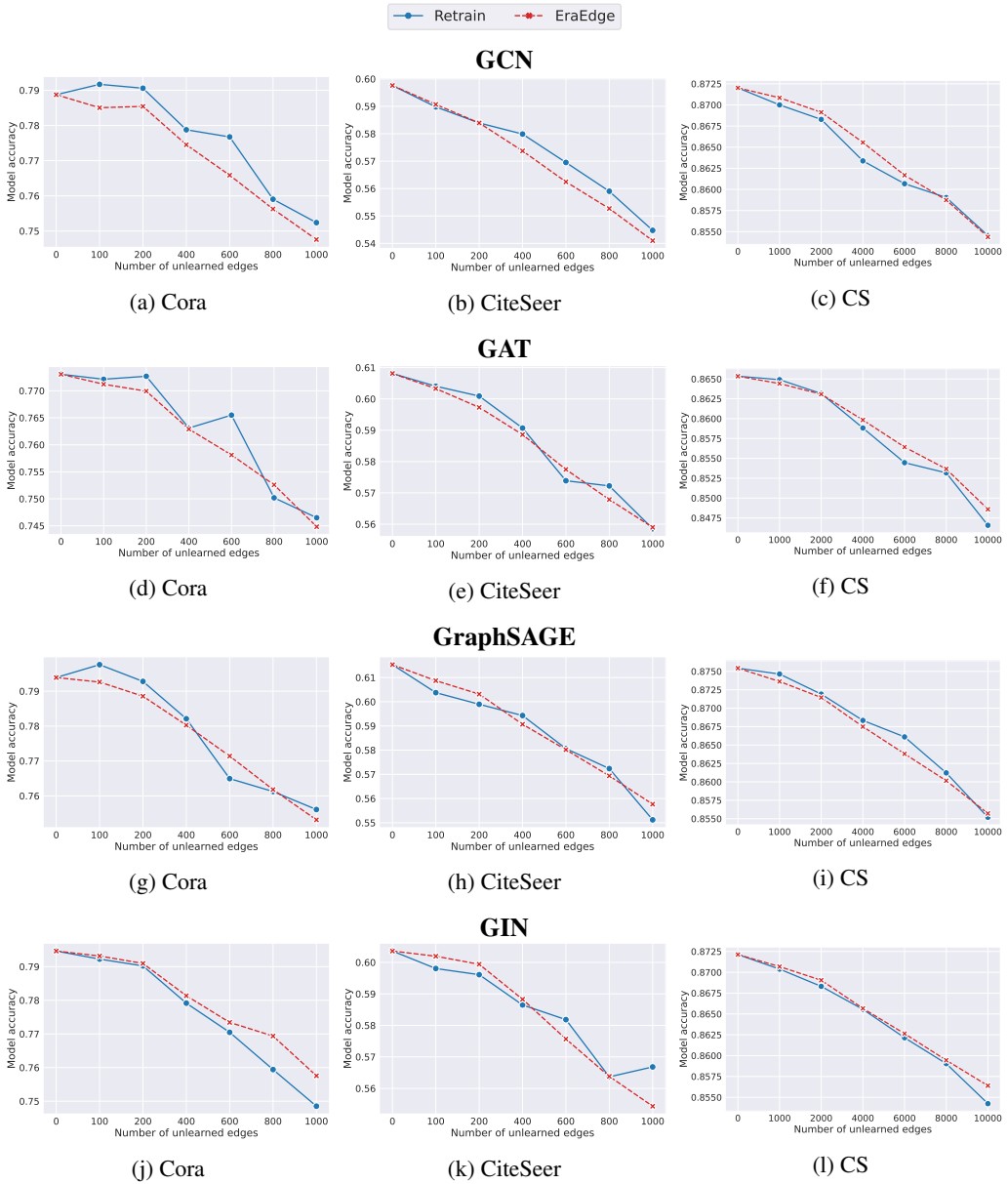

Figure 5: **Accuracy of the target GNN model after retraining/unlearning**. The results show that EraEdge still maintains a competitive model accuracy which is close to that of the retraining model.

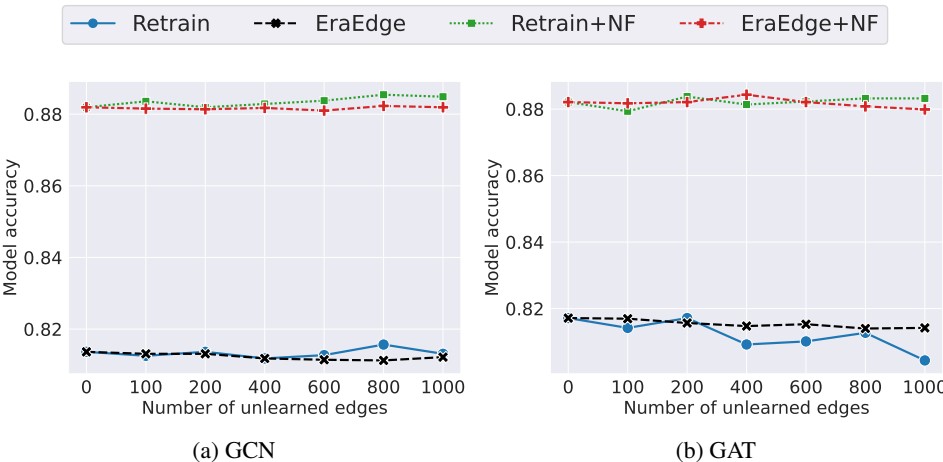

Figure 6: Comparison of target model accuracy for unlearning with and without the original node features (Cora). "XX+NF" indicates the model that considers the original node features.

mance of EraEdge that uses the original node features. Figure 6 reports the target model accuracy of the unlearned model that is trained with or without the node features. We have the following main observations. First, the target model accuracy improves significantly by considering the original node features. This shows that the node features have dominant importance on the target model performance in this setting. However, the target model accuracy of the unlearned model still stays close to that of the retrained model. In other words, EraEdge still can make GNNs forget the edges effectively even when node features have dominant importance over the graph structure on model performance.

