# OpenReview forum: "Fast Yet Effective Graph Unlearning through Influence Analysis"
_ICLR.cc/2023/Conference — Submitted to ICLR 2023_

### Official Review · Reviewer_UL7T · 2022-10-19

**Confidence:** 5
**Correctness:** 1
**Technical Novelty And Significance:** 2
**Empirical Novelty And Significance:** 2
**Recommendation:** 3

**Clarity, Quality, Novelty And Reproducibility:**

- Clarity: The authors should specify the difference between exact and approximate unlearning. Note that their main baseline methods such as Retrain and GraphEraser are exact unlearning methods while the proposed method is an approximate unlearning method without theoretical guarantees. Hence, merely comparing the test accuracy and time complexity is not satisfied.

- Quality: I have concerns about the privacy of EraEdge method.

- Novelty: Most of the analysis and techniques are from the existing literature. The main novelty comes from applying them to graphs, yet the novelty of this extension used in the papers seems limited.

- Reproducibility: The authors do not provide their experimental code. Details about hyperparameters are also missing.


**Strength And Weaknesses:**

### Strengths

- EraEdge can be applied to non-linear models such as GNNs.

- The methods seem to be efficient in terms of time complexity.

- The problem of machine unlearning on graphs is an important problem.

### Weaknesses

- The proposed unlearning definition is not rigorous and heuristic-based. Also, having approximately close model output does not guarantee privacy.

- The proposed method is an ***approximate unlearning*** method ***without theoretical guarantee*** (i.e. heuristic based).

- The related works about machine unlearning on graphs are not extensive enough. Also, differential privacy based GNNs should also be discussed.

- In the experiment, the authors use random Gaussian vectors as node features instead of the default features. How does the result look like if we use the default features?

- (Minor) The paper focuses on the edge unlearning problem, while the really important problem is the node unlearning problem.

### Detail comments

While the problem of machine unlearning on graphs is very important, one should be extremely careful when they claim a method can achieve unlearning. Machine unlearning can be roughly characterized into two categories, exact and approximate unlearning. For exact unlearning methods, we require the unlearned models to be ***identical*** (in distribution) to the one retraining from scratch, examples include sharding-based method as mentioned by the authors. Approximate unlearning requires the unlearned model to be *indistinguishable* from the one retraining from scratch. Note that one should be extremely careful when defining the *indistinguishability*, as an inappropriate definition could lead to zero privacy in some cases (see [1] for a simple example). Apparently, the proposed definition of unlearning belongs to approximate unlearning and the authors should clearly specify it. Otherwise, the paper can be misleading to the readers.

One rigorous definition of approximate unlearning is via differential privacy type of definition in the parameter space, which was proposed in [1]. The proposed approximate unlearning method therein comes with a differential privacy type of theoretical guarantee. The other line of work such as [2] proposed a heuristic-based measure, which is more similar to the one proposed in this paper. However, the proposed method in [2] involves ***privacy noise*** to *blur out* the potentially leaked information. In contrast, EraEdge does not add any privacy noise to protect the privacy. Most importantly, the authors of [2] conduct extensive experiments and multiple metrics to examine the effectiveness of unlearning their method. On the other hand, the authors of this paper merely examine the closeness of the model output which can be problematic following the same rationale in the counterexample provided in [1] (i.e. the paragraph in the title *Insufficiency of parametric indistinguishability*). Hence, I doubt how private it is for EraEdge on the unlearned data. I would suggest the authors conduct similar extensive experiments to [2], or examine the ability of the proposed method against some membership inference attack methods.

Regarding the related works, the authors miss several recent papers about machine unlearning on graphs. Please check the survey paper [3] for a collection of papers about machine unlearning on graphs, where the node unlearning problem is also studied. It is also worth mentioning the recent development of differential private GNNs, as differential private models automatically achieve (approximate) unlearning *without* any update [1].

In summary, my main concern about the paper is the privacy of the proposed method. Since EraEdge is not an exact unlearning approach, one has to be extremely careful about the choice of *indistinguishability*. Also, most of the existing approximate machine unlearning methods require adding privacy noise to burr out the information from the approximation error. However, EraEdge does not leverage any privacy noise and thus any approximate error can potentially lead to severe privacy issues (i.e. adversarial attacks). In the empirical evaluation, the authors do not conduct enough experiments to demonstrate the privacy of their EraEdge. This should be done for heuristic-based methods such as those in [2]. Hence, I feel the paper needs a major revision before publishing.

### References

[1] Certified data removal from machine learning models, Guo et al., ICML 2020.

[2] Eternal Sunshine of the Spotless Net: Selective Forgetting in Deep Networks, Golatkar et al., CVPR 2020.

[3] A Survey of Machine Unlearning, Nguyen et al., 2022.


**Summary Of The Paper:**

The authors study the machine unlearning problem on graphs, where they focus on the edge unlearning problem. The proposed method, EraEdge, is based on subtracting the *influence* of the unlearned edges in a heuristic manner. The experiments demonstrate the efficiency and accuracy of the proposed method. They also show that the output of the unlearned model is close to the one retraining from scratch.

**Summary Of The Review:**

My major concern focuses on how private the proposed unlearning method is. I also think the authors should specify the difference between exact and approximate unlearning, otherwise the result can be misleading. For example, EraEdge is an approximate unlearning method while retraining from scratch is an exact unlearning method. Once we specify this difference, one can naturally ask how EraEdge *trade-off* privacy for time complexity and test accuracy. I also find the experiment setting a bit weird, where the authors replace the default node features with random Gaussian vectors. This is an artificial setting, and I wonder how the result looks like if the authors use default node features. In summary, I feel the paper needs a major revision before publishing.

---

> ### Author Response · Authors · 2022-11-11
> **Response to Reviewer UL7T**
>
> Thanks for your valuable comments. Please see below for our point-to-point response.
>
> **Question 1 [Evaluation of edge forgetting ability of the proposed method]**
>
> **Response:** Thanks for the suggestion. We added new experiments that measure the privacy vulnerability of the removed edges by the membership inference attack against the unlearned model. The details of the attack and the performance results can be found in Section 5.3. Our main observation is that, while the membership inference attack is effective to infer the edges to be removed from the original model, its ability of inferring these edges from either the retrained or the unlearned model degrades. Furthermore, the extent to which EraEdge forgets the edges is similar to that of the retrained model.
>
> We also thank the reviewer for the suggestion of comparing with the existing work of certified machine unlearning by adding privacy noise. We added the new experimental results that use the certified graph unlearning (CGU) [1] as a baseline. The results show that, compared with CGU, our method provides much higher target model accuracy and comparable privacy, where privacy is evaluated as the privacy risk of the removed edges against the membership inference attack. In other words, our unlearning solution better addresses the trade-off between model accuracy and privacy. The details of the results can be found in Section 5.4.
>
> [1] Chien, Eli, Chao Pan, and Olgica Milenkovic. "Certified graph unlearning." NeurIPS 2022 New Frontiers in Graph Learning Workshop (NeurIPS GLFrontiers 2022).
>
> **Question 2 [Missing related work]**
>
> **Response:** Thanks for bringing these papers to our attention. We have added the existing work on certified machine learning into Related Work in the revision. We also checked the graph unlearning papers in the survey, and found that the two papers ([28,29] in the survey) have not been published yet. Thus we will not include these two papers in our manuscript at present.
>
> **Question 3 [Using default features]**
>
> **Response:** We added new experimental results that measure the model accuracy of using default features. The results can be found in Appendix A.5. The main observation is that our unlearning method still can make GNNs forget the edges effectively even when node features have dominant importance over the graph structure on model performance.

---

> > ### Comment · Reviewer_UL7T · 2022-11-16
> > **My main concerns remain and new concerns arise pertaining to new experiments**
> >
> > I thank the authors for their effort in addressing my concerns. I do appreciate the new membership inference attack results and the comparison to CGU [3]. However, my main concerns remain or are unaddressed, and I have questions regarding the new experimental results.
> >
> > ### Does EraEdge really ***private***?
> >
> > Note that the authors do not leverage the common definition of privacy as in the differential privacy literature. They define their privacy based on the ***closeness*** of model output, where the gold standard is the retrained model. They try to minimize the ``distance'' of their model output to the retrained model, where the distance is chosen as JSD in the experiment. Note that their method EraEdge does not include any kind of ***privacy noise***.
> >
> > As also mentioned by the Reviewer TMqC, this privacy noise is crucial and ubiquitous in many previous works on privacy, including CGU. The main reason that we need it is that we want to prevent the obvious worst-case example mentioned in [1]. I already mentioned this counterexample in my original review but the authors do not respond to it. To make my point more clear, I state the counterexample stated in [1] as follows.
> >
> > Consider a linear regressor trained on the dataset $\mathcal{D} = \lbrace \( \mathbf{e}_i , i \) \rbrace _{i \in [d]}$, where $\mathbf{e}_i$'s are the standard basis vector in $\mathbb{R}^d$. A regressor that is initialized with zeros will place a non-zero weight on $\mathbf{w}_i$ if $(\mathbf{e}_i,i)$ is included in $\mathcal{D}$, and a zero weight on $\mathbf{w}_i$ if not.
> >
> > Clearly, an approximate removal mechanism that leaves $\mathbf{w}_i$ small but non-zero still reveals that $(\mathbf{e}_i,i)$ appeared during training (actually, no privacy at all!). Similarly, classification problems with similar settings (i.e., changing $(\mathbf{e}_i,i)$ to $(\mathbf{e}_i,\mathbf{e}_i)$) suffer from the same problem, no matter we examine the weights or the output. With privacy noise, one can prevent this simple worst case and the DP-type guarantee further characterize *how private our data is*. This is why even a heuristic-based method such as that in [2] still adds noise. Thus, I still think the methodology proposed by the authors cannot guarantee ***privacy*** (with a definition that normal users would accept). I think it is quite fair for everyone to ask *how private is EraEdge* and what the ***privacy*** defined therein can guarantee to the data providers.
> >
> > ### Comparison to CGU seems problematic
> >
> > It is great to see that the authors attempt to compare their method with CGU under several criteria. However, I find the experiment results and setting seem problematic and inconsistent with those reported in the CGU paper. The authors of CGU paper [3] choose the standard deviation of the Gaussian noise to be $0.1$ achieving $(1,10^{-4})$-certified removal. In contrast, the authors of EraEdge set the standard deviation to be $1$ which is unnecessarily high. We can see that CGU is reported to achieve above $80$\% in accuracy for edge unlearning [3] while the authors of EraEdge report a below $50$\% accuracy for CGU. I hope the authors can explain why they choose such an unnecessarily large noise for CGU, given the fact that using merely a standard deviation of $0.1$ can ensure quite good privacy. Again, since the authors neither provide a theoretical guarantee for their EraEdge nor provide their code and implementation, it is hard for me to judge the correctness of their experimental results.
> >
> > Please do let me know if I misinterpret something or if there are problems with my comments.
> >
> > ### References
> > [1] Certified data removal from machine learning models, Guo et al., ICML 2020.
> >
> > [2] Eternal Sunshine of the Spotless Net: Selective Forgetting in Deep Networks, Golatkar et al., CVPR 2020.
> >
> > [3] Chien, Eli, Chao Pan, and Olgica Milenkovic. "Certified graph unlearning." NeurIPS 2022 New Frontiers in Graph Learning Workshop (NeurIPS GLFrontiers 2022).

---

### Official Review · Reviewer_wUNT · 2022-10-22

**Confidence:** 3
**Correctness:** 3
**Technical Novelty And Significance:** 2
**Empirical Novelty And Significance:** 2
**Recommendation:** 5

**Clarity, Quality, Novelty And Reproducibility:**

I think I have made these points clear in the above response.

Clarity: Generally easy to follow while some explanations on experiment settings are missed

Quality: Okay, but not extensive enough given previous works

Novelty: Topic is interesting while there are a few contributions being over-claimed.

Reproducibility: Good while I did not see how to tune hyperparameters and some detailed experiment settings are missed.

**Strength And Weaknesses:**

Strengthes:
1. Graph unlearning is a relatively novel concept. The problem studies here is interesting. Also, the analysis and argument sound reasonable and solid.

2. The paper is written very well. I appreciate the logic flow. The motivation and the exposition of the approach is clear.

Weaknesses:
1. Here is my biggest concern. Although I overall think the technique in this paper is reasonable and the studied problem is interesting. Unfortunately, recently, I have read a relevant paper on graph unlearning published four months ago [1], which I think has studies a far more extensive setting on graph unlearning than this work. Although that work is just an arxiv paper, I cannot view it as a concurrent work because the content studied in [1], in my opinion, is broader and provides more insights than the setting studied in this work. I know it is tough for the authors but I cannot ignore this. Therefore, this work's statement saying this is the only work that considers unlearning in GNN, which is an over-claiming.

Moreover, [1] studies both node and edge unlearning, while this work only studies edge unlearning. In my opinion, node unlearning is more crucial because a user (typically corresponding to a node), if not wanting to disclose her data, will ask to remove this node from the graph. Moreover, I think [1] also tells more data insights due to their analytic bounds such as the dependence of unlearning performance on node degrees, etc.

I can see some adopted detailed techniques are different, such as [1] using SGC (linear model) while this paper using convex assumptions. My feeling is that if this work may discuss both edge/node unlearning and also provides further insights on how graph structure affects the unlearning performance, I would appreciate the technical difference in this paper and may support an acceptance. However, the current setting of this paper is still kind of narrow compared to [1].

2. This work writes well and is easy to follow. However, I feel there is a little bit misleading in the introduction. For example, after reading the introduction, I thought the paper would like to touch non-convex settings in theory (the fourth paragraph of intro). However, the later analysis is based on convex assumption. Moreover, in intro, the authors say "empirical studies on tradeoff between unlearning efficiency, accuracy, unlearning efficacy". However, in the experiments, I can only see the list of these results without tradeoff. My understanding of the tradeoff would be about, e.g., high accuracy/efficacy  requires less efficiency, and the proposed method has a work to balance these aspects. Unfortunately, I do not think the proposed approach has such flexibility.

3. Moreover, I am not clear how the averaged JSD is computed. I can think of there are multiple ways to define averaged JSD. Do you mean averaging over testing samples? or averaging over classes? I think a math equation is needed to show this.

4. Fig.4 the first row has wrong subtitles I think.

5. Regarding experiments, how do you remove edges, randomly and how many times, or? Also, I do not see how to tune the model and how to make sure the comparison between model retraining and the proposed method fair, e.g., same model size? how about learning rate?

6. Since efficiency is one topic of interest in this work, the used graphs are in general too small.



[1] Certified Graph Unlearning, Chien et al., 2022.

**Summary Of The Paper:**

This work studies a notion of unlearning for graph neural networks. Basically, given a set of edges removed from the graph, it tries to address how to fast adjust the model parameters to make the model behave like the model retrained on the graph with edge removal while without retraining. The technical idea is to analyze the influence of the edge removal on the model parameters given the convex and differentiable assumption of the objective. Experiments show some superiority of the proposed method.

**Summary Of The Review:**

The studied problem is interesting. The adopted technique is reasonable. However, the novelty and true technical contributions are not sufficient given previous works.

---

> ### Author Response · Authors · 2022-11-11
> **Response to Reviewer wUNT**
>
> Thanks for your valuable comments. Please see below for our point-to-point response.
>
> **Question 1.a [Comparison with certified graph unlearning work in literature]:**
>
> **Response:** Thank you for bringing the paper of certified graph unlearning (CGU) to our attention. We added new experimental results that use the certified graph unlearning (CGU) work as a baseline. The results show that, compared with CGU, our method provides much higher target model accuracy and comparable privacy, where privacy is evaluated as the privacy risk of the removed edges against the membership inference attack. In other words, our unlearning solution better addresses the trade-off between model accuracy and privacy. The details of the results can be found in Section 5.4.
>
> **Question 1.b [Node unlearning]:**
>
> **Response:** We agree that node unlearning is also an important topic. However, we believe that our edge unlearning solution can be easily extended to handle node unlearning. The discussion of the possible extension can be found in Section 7.
>
> **Question 2 [Claim of contribution on non-convex setting in Introduction]:**
>
> **Response:** Unfortunately, theoretical analysis for non-convex settings remains open. However, in section 4, we discussed the convergence guarantee for non-convex settings. Also, our empirical results demonstrate that EraEdge works well even for non-convex settings.
>
> **Question 3 [Calculation of average JSD ]:**
>
> **Response:** We compute the average of JSD of posteriors between retrained models and unlearned models over all test samples. Formally, $AJSD$ = $\frac{1}{|V_{test}|}\sum_{v\in V_{test}}JSD(p_v, p_v')$, where $|V_{test}|$ denotes the number of nodes in the testing data, and $p$ and $p'$ denote the posterior output of node v by the retrained and unlearned models, respectively.
>
> **Question 4 [Wrong subtitles for 1st row of Fig.4 ]:**
>
> **Response:** We have fixed it in the revision (now it is Figure 5).
>
> **Question 5 [How do you remove edges, randomly and how many times, or? Also, I do not see how to tune the model and how to make sure the comparison between model retraining and the proposed method fair, e.g., same model size? how about learning rate?]:**
>
> **Response:** We randomly pick $k$ =\{100, 200, 400, 600, 800, and 1,000\}) edges from Cora and CiteSeer datasets, and $k$=\{1,000, 2,000, 4,000, 6,000, 8,000, and 1,0000\}) edges from CS dataset for removal. We pick more edges from the CS dataset  as it is much larger than the Cora and Citeseer datasets. For each setting, we randomly sample 10 batches with each batch of $k$ edges, and take the average of model performance (model accuracy, unlearning efficacy, etc.) of the 10 batches.
>
> To ensure fair comparison between the retrained and unlearned models, we use the same model size (i.e., same number of layers and number of neurons) for both retraining and unlearned models. All GNN models are trained with a learning rate of 0.001. We train the models by 1,000 epochs, with the early-stopping condition as that the validation loss does not decrease for 20 epochs. We have added these details into Appendix A.2 in the revision.
>
> **Question 6 [Graphs in experiments are too small]:**
>
> **Response:** The CS dataset that we used in the experiments is relatively large (~18K nodes and 163K edges). We have shown the results of unlearning efficiency on this large graph in Table 2 and Figure 4.

---

> ### Comment · Reviewer_wUNT · 2022-11-23
> **Thanks!**
>
> Thank you so much for the response. The authors have addressed most of my concerns but not all of them. The weakness (1) about less extensiveness than [1] was not addressed from my perspective. But I can increase my evaluation from 3 to 5 to acknowledge the revision.
>
> [1] Chien, Eli, Chao Pan, and Olgica Milenkovic. "Certified graph unlearning."

---

### Official Review · Reviewer_yNN2 · 2022-10-25

**Confidence:** 4
**Clarity, Quality, Novelty And Reproducibility:** It is understandable to a large exten…
**Correctness:** 2
**Technical Novelty And Significance:** 2
**Empirical Novelty And Significance:** 2
**Recommendation:** 3

**Strength And Weaknesses:**

Strength

1. The proposed method has a good applicability and can work with most existing variants of GNNs.

2. The paper is a computationally and memory efficient algorithm.

3. The algorithm is evaluated on a number of three real datasets to demonstrate the effectiveness of the proposed approach for graph unlearning.

Weakness

1. Upweighting using the influence function merely approximates the retrained parameters and provides no theoretical error bound on non-convex loss functions. This work then constructs another approximation of this already fuzzy target to obtain the unlearnt model. One can hardly be convinced that such a method will result in a set of parameters that resemble the retrained model. The proposed scheme also provides no means of generating a verification of data removal, which is utterly vital for a data provider.

2. Adding a scaled identity matrix to the hessian to make it positive definite only accounts for the non-invertible problem. It still destroys the basis of Theorem 2, which requires the loss function to be globally convex. This assumption is too strong to make any meaningful sense in real-world scenarios. Also, Theorem 2 seems to be only an adaptation in notations of eq.(3) in [1].

3. In scenarios where the graph is relatively dense and the GCN is deep (e.g., as described in [2]), the affected set of nodes, as defined in the paper, can easily become the whole graph, which sort of destroys the purpose of the paper, which is fast unlearning.

4. Unlearning efficacy is only evaluated for the proposed model without comparison with other baselines, unlike classification accuracy. Also, experiments like the behavior difference between the unlearning model and the retrained model on the unlearned part of the dataset should be added to demonstrate the algorithm's efficacy further.

[1] Pang Wei Koh and Percy Liang. “Understanding Black-box Predictions via Influence Functions.” In: Proceedings of the 34th International Conference on Machine Learning, ICML 2017, Sydney, NSW, Australia, 6-11 August 2017. Ed. by Doina Precup and Yee Whye Teh. Vol. 70. Proceedings of Machine Learning Research. PMLR, 2017, pp. 1885–1894.

[2] Guohao Li et al. “DeepGCNs: Can GCNs Go As Deep As CNNs?” In: 2019 IEEE/CVF International Conference on Computer Vision, ICCV 2019, Seoul, Korea (South), October 27 - November 2, 2019. IEEE, 2019, pp. 9266–9275.

**Summary Of The Paper:**

An unlearning algorithm for graph neural networks is proposed in the paper. The paper tries to find the difference between the upweighted model and the original model by utilizing a hessian-based approximation, which can be solved by investigating the corresponding linear system with conjugate gradient methods. The found difference is then added to the original model to obtain the retrained parameters.

**Summary Of The Review:**

In general, the studied problem is interesting and important. In addition, the methodology is principled with three major merits as discussed above. However, the work still has some unaddressed concerns to well justify its technical and empirical contributions.

---

> ### Author Response · Authors · 2022-11-11
> **Response to Reviewer yNN2**
>
> We thank the reviewer for their valuable feedback on our manuscript. We are happy that the reviewer finds the algorithm to be computationally and memory efficient.
>
> **Question 1 [Definition of unlearning and verification of removal effects]:**
>
> **Response:** First, we want to clarify that our unlearning notion is not defined as the indistinguishability of model parameters between the restrained and unlearned models. Instead, we require the indistinguishability of model outputs (e.g., probability distribution) between the retrained and unlearned models. This unlearning definition has been considered in the literature (e.g.,[1,2]). We agree that EraEdge does not provide any theoretical error bound. Indeed, the inference of the error bound on the model outputs is very challenging. Therefore, we perform the empirical studies to demonstrate the closeness of the retrained and unlearned model in the model output space.
>
> [1] Golatkar, Aditya, Alessandro Achille, and Stefano Soatto. "Eternal sunshine of the spotless net: Selective forgetting in deep networks." Proceedings of the IEEE/CVF Conference on Computer Vision and Pattern Recognition. 2020
>
> [2] Thudi, Anvith, et al. "On the necessity of auditable algorithmic definitions for machine unlearning." 31st USENIX Security Symposium (USENIX Security 22). 2022.
>
> In terms of verification of removal effects, we added new experiments that launch the membership inference attack (MIA) against the unlearned models to predict the existence of the removed edges in the training graph. The results can be found in Section 5.3. Our main observation is that, while MIA is effective to infer the edges to be removed from the original model, its ability of inferring these edges from either the retrained or the unlearned model degrades to around random guess. Furthermore, the extent to which EraEdge forgets the removed edges is similar to that of the retrained model. This demonstrates the effectiveness of EraEdge in removing the effect of edges from the GNN models.
>
> **Question 2 [Convex assumption of Theorem 2]:** *Adding a scaled identity matrix to the hessian to make it positive definite only accounts for the non-invertible problem. It still destroys the basis of Theorem 2, which requires the loss function to be globally convex. This assumption is too strong to make any meaningful sense in real-world scenarios. Also, Theorem 2 seems to be only an adaptation in notations of eq.(3) in [1].*
>
> **Response:** If the magnitude of the negative eigenvalues of the Hessian matrix is not large, then adding a scaled identity matrix will turn the problem into convex, and hence Theorem 2 applies. Unfortunately, for the broad problem of optimizing graph neural networks, it remains largely open to come up with convergence guarantee on the global optimization landscape. This is out of the scope of this work.
>
> **Question 3 [Evaluation of unlearning efficacy]:**
>
> **Response:** Please see our response to Question 1  [Definition of unlearning and verification of removal effects] above, in particular the part of the new experiments of MIA evaluation.

---

### Official Review · Reviewer_TMqC · 2022-10-25

**Confidence:** 4
**Clarity, Quality, Novelty And Reproducibility:** The paper is overall novel and easy t…
**Correctness:** 3
**Technical Novelty And Significance:** 3
**Empirical Novelty And Significance:** 3
**Recommendation:** 5

**Strength And Weaknesses:**

Strengths:
- Unlearning on GNNs is less studied, and there is no existing works on influence function-based GNN unlearning.
- The paper is overall easy to follow.
- The experimental results demonstrate that EraEdge can efficiently unlearn a set of edges via the indistinguishability between the retrained model parameters and the unlearned model parameters.

Concerns/Questions:
- Many existing influence function-based approximate unlearning techniques (see references below) add Gaussian noises to preserve data privacy. I am wondering why such noise is not needed in EraEdge for data privacy?
[1] https://arxiv.org/abs/1911.04933
[2] https://arxiv.org/abs/1911.03030
[3] https://arxiv.org/abs/2106.04378
[4] https://arxiv.org/abs/2006.14755
[5] https://arxiv.org/abs/2007.02923
[6] https://arxiv.org/abs/2012.13431
[7] https://arxiv.org/abs/2110.11891
[8] https://arxiv.org/abs/2103.03279
- Existing works often solve Eq. (10)-like equation with Hessian-vector product (HVP). What is the key limitation of using HVP to solve Eq. (10)? If HVP can be used as well, what is the benefit of conjugate gradient over HVP?
- I doubt that the indistinguishability in the GNN output necessarily mean the data is removed. Similar concern also appear in (Guo et al. 2020) even for linear and convex classifier like logistic regression, not to mention the nonconvexity of GNNs. It is better to study the privacy of this data removal mechanism like how they are studied in differential privacy (e.g., through membership inference attack).
- How would EraEdge perform when we sequantially delete multiple batches of edges? In many real-world scenarios, it is uncommon that the model will be only unlearned once.

**Summary Of The Paper:**

This paper studies machine unlearning on graph neural networks (GNNs) by analyzing influence function. The authors identify that simply applying influence function on GNNs for edge removal is problematic due to node dependency. As such, the authors propose to estimate the influence function by upweighting the set of all affected nodes. Then the influence is obtained as the reverse of Hessian matrix multiplied by the gradient vector. And conjugate gradient is applied to reduce the computational cost. Experimental results demonstrate the effectiveness of the proposed EraEdge in terms of the indistinguishability betweem model parameters and the efficiency of the proposed EraEdge in terms of running time.

**Summary Of The Review:**

While the paper is novel in terms of its technical aspect, I have a few concerns on the technical details on the privacy of this unlearning approach. In terms of writing, the paper is clear and easy to follow.

---

> ### Author Response · Authors · 2022-11-11
> **Response to Reviewer TMqC**
>
> We thank the reviewer for your nice summary of our contributions and insightful comments. Below is our point-to-point response to your questions.
>
> **Question 1 [Comparison with existing methods that adds noise to unlearning]** *Why noise is not needed in EraEdge for data privacy?*
>
> **Response:** We agree that noise can be added during unlearning to protect data privacy, as some prior work did. However, adding noise can lead to significant accuracy loss of the target model. To measure the impact of noise on unlearning, we consider *certified graph unlearning* (CGU) [1] as a baseline, which adds noise to the loss function. We compare its performance with our EraEdge method in terms of target model accuracy, unlearning efficacy, and privacy vulnerability of the removed edges against the membership inference attack. The new results can be found in Section 5.4 in the revision. Our major observation is that, compared with CGU, our method provides much higher target model accuracy and comparable privacy which is evaluated as the privacy risk of the removed edges against the membership inference attack. In other words, our unlearning solution better addresses the trade-off between model accuracy and privacy.
>
> [1] Chien, Eli, Chao Pan, and Olgica Milenkovic. "Certified graph unlearning." NeurIPS 2022 New Frontiers in Graph Learning Workshop (NeurIPS GLFrontiers 2022).
>
> **Question 2 [Clarification of HVP]:** *What is the key limitation of using HVP to solve Eq. (10)? If HVP can be used as well, what is the benefit of conjugate gradient over HVP?*
>
> **Response:** There are practical challenges to directly apply HVP to solve Eq. (10). HVP aims to efficiently solve the product of a Hessian with a certain vector. However, it is hard to solve the product of **the inverse of Hessian** with a certain vector. That is why we leveraged a conjugate gradient solver.
>
> **Question 3 [Evaluation of unlearning effect]:** *It is better to study the privacy of this data removal mechanism (e.g., through membership inference attack).*
>
> **Response:** We agree that simply measuring the indistinguishability in GNN outputs does not necessarily mean that the edges are removed. We added new experiments that measure the privacy vulnerability of the removed edges by the membership inference attack against the unlearned model. The details of the attack and the performance results can be found in Section 5.3. As we can see, while the membership inference attack is effective to infer the edges to be removed from the original model, its ability of inferring these edges from either the retrained or the unlearned model degrades. Furthermore, the extent to which EraEdge forgets the edges is similar to that of the retrained model.
>
> **Question 4 [Sequential unlearning]:**  *How would EraEdge perform when we sequentially delete multiple batches of edges*
>
> **Response:** We added new empirical studies of sequential unlearning (i.e., multiple batch deletion requests to forget the edges in a sequential fashion), and compared the target model accuracy of sequential unlearning with one-batch unlearning. The results can be found in Appendix A.4.  Our main observation is that the accuracy of the unlearned model remains close to the retrained model at each batch during sequential removals. Furthermore, the performance of the unlearned model after sequential removal stays close to that of the model after single-batch unlearning. These results demonstrate that EraEdge can handle sequential deletion effectively.

---

> > ### Comment · Reviewer_TMqC · 2022-11-19
> > **Feedback**
> >
> > Thank you for your efforts in the rebuttal phase. I am still not convinced by not including privacy noise. I think there are more justifications needed as previous works have demonstrate the importance of privacy risks rigorously.
> >
> > In addition, in [1], the authors use HVP to calculate the inverse of Hessian times a vector. They also consider CG as a potential solution and compared CG with HVP. Based on their claim, HVP is a better choice than CG. So more clarifications are needed.
> >
> > [1] Koh, P. W., & Liang, P. (2017, July). Understanding black-box predictions via influence functions. In International conference on machine learning (pp. 1885-1894). PMLR.

---

### Decision · Program_Chairs · 2023-01-20

**Decision:**

Reject

**Justification For Why Not Higher Score:**

This paper does not reach the bar of ICLR, which lacks clear motivations and suffers from some technical flaws. More details can be found in my meta review.

**Justification For Why Not Lower Score:**

N/A

**Metareview: Summary, Strengths And Weaknesses:**

Based on the collected information from all of reviewers and my personal judgment, I can make the recommendation on this paper, **rejection**. Here are the comments that I summarized, which include my opinion and evidence.

**Research Problem**

The authors consider the graph unlearning problem, which can be regarded as a novel but less explored problem.

**Presentation**

This paper is easy to read, but difficult to follow. I read this paper more than three times and found this paper is less coherent. I have the same feeling with Reviewer wUNT. In the abstract, the authors mentioned the challenges in non-convexity nature of GNNs and large scale of input graph. Unfortunately, this paper does not tackle the above two points.

**Challenge Analysis or Motivation**

The challenge analysis is missing. I did not find anything about it in the original version. In the updated version, the authors mention they aim to design the efficient approximate unlearning solutions that are model-agnostic. I believe the proposed technique struggles to handle generalized PageRank.

**Philosophy**

Without a clear challenge analysis, it is unclear the philosophy to deal with the challenges.

**Technical Contribution**

(1) The authors extend Koh & Liang’s work from the tabular data to graph data without rigorous support, especially in the non-convex scenarios. When dealing with non-convexity of GNNs, it is necessary to verify such approximations are accurate enough. Lambda is a key parameter to guarantee the existence of the Hessian matrix and determine the final result; however, I do not see parameter analysis on lambda or how to set it. Moreover, removing a set of edges would bring extra group effect as well [1]. The authors need to address this issue in this scenario as well. From Figure 2, we can see the large JSD value with the increased unlearned edges.

(2) Another technical contribution claimed by the authors is that they resolve the computational challenge on calculating the inverse of a large Hessian matrix. Unfortunately, the techniques on Page 6 are well-known, at least known to me. The authors also cited the original papers as well. Therefore, I do not regard this as a contribution point.

(3) Two reviewers raise their concerns on privacy and provide a list of key references as well. However, privacy is not covered in the main body, but indeed it is a crucial topic in the unlearning area. Later, the authors added one more experiment on this point. I suggest the authors to add more discussions on why the proposed method can address this problem.

**Experiments**

(1) The parameter analysis on lambda is missing.

(2) Figure 2 does not include the results from BLPA and BEKM.

(3) I believe Reviewer wUNT would like to see some large graphs, such as Ogbn-Arixv that contains nearly 170 thousand nodes and over 2 million edges.

No objection was raised from the reviewers on the rejection recommendation.

[1] Koh, P. W. W., Ang, K. S., Teo, H., & Liang, P. S. (2019). On the accuracy of influence functions for measuring group effects. Advances in neural information processing systems, 32.

**Summary Of Ac-Reviewer Meeting:**

This is not a borderline paper.